# WINNING THE LOTTERY WITH CONTINUOUS SPARSIFICATION

## ABSTRACT

The Lottery Ticket Hypothesis from Frankle & Carbin (2019) conjectures that, for typically-sized neural networks, it is possible to find small sub-networks which train faster and yield superior performance than their original counterparts. The proposed algorithm to search for "winning tickets", Iterative Magnitude Pruning, consistently finds sub-networks with $90 - 95\%$ less parameters which train faster and better than the overparameterized models they were extracted from, creating potential applications to problems such as transfer learning.

In this paper, we propose Continuous Sparsification, a new algorithm to search for winning tickets which continuously removes parameters from a network during training, and learns the sub-network's structure with gradient-based methods instead of relying on pruning strategies. We show empirically that our method is capable of finding tickets that outperforms the ones learned by Iterative Magnitude Pruning, and at the same time providing faster search, when measured in number of training epochs or wall-clock time.

## 1 INTRODUCTION

Although deep neural networks have become ubiquitous in fields such as computer vision and natural language processing, extreme overparameterization is typically required to achieve state-of-the-art results (Xie et al., 2017; Devlin et al., 2018), causing higher training costs and hindering applications where memory or inference time are constrained. Recent theoretical work suggest that overparameterization plays a key role in both the capacity and generalization of a network (Neyshabur et al., 2018), and in training dynamics (Allen-Zhu et al., 2019). However, it remains unclear whether overparameterization is truly necessary to train networks to state-of-the-art performance.

At the same time, empirical approaches have been successful in finding less overparameterized neural networks, either by reducing the network after training (Han et al., 2015; 2016) or through more efficient architectures that can be trained from scratch (Iandola et al., 2016). Recently, the combination of these two approaches lead to new methods which discover efficient architectures through optimization instead of design (Liu et al., 2019; Savarese & Maire, 2019). Nonetheless, parameter efficiency is typically maximized by pruning an already trained network.

The fact that pruned networks are hard to train from scratch (Han et al., 2015; 2016) suggests that, while overparameterization is not necessary for a model's capacity, it might be required for successful network training. Recently, this idea has been put into question by Frankle & Carbin (2019), where heavily pruned networks are trained faster than their original counterparts, often yielding superior performance.

A key finding is that the same parameter initialization should be used when re-training the pruned network. A *winning ticket*, defined by a sub-network and a setting of randomly-initialized parameters, is quickly trainable and has already found applications in, for example, transfer learning (Morcos et al., 2019; Mehta, 2019; Soelen & Sheppard, 2019), making the search for winning tickets a problem of independent interest.

Currently, the standard algorithm to find winning tickets is Iterative Magnitude Pruning (IMP) (Frankle & Carbin, 2019; Frankle et al., 2019), which consists of a repeating a 2-stage procedure that alternates between parameter optimization and pruning. As a result, IMP relies on a sensible choice for pruning strategy, and is time-consuming: finding a winning ticket with $1\%$ of the original parameters in a 6-layer CNN requires over 20 rounds of training followed by pruning, totalling over 1000 epochs (Frankle & Carbin, 2019). Choosing a parameter's magnitude as pruning criterion

has also shown to be sub-optimal in some settings (Zhou et al., 2019), leading to the question of whether better winning tickets can be found by different pruning methods. Moreover, at each iteration, IMP resets the parameters of the network back to initialization, hence considerable time is spent on re-training similar networks with different sparsities.

With the goal of speeding up the search for winning tickets in deep neural networks, we design a novel method, Continuous Sparsification, which continuously removes weights from a network during training, instead of following a strategy to prune parameters at discrete time intervals. Unlike IMP, our method approaches the search for sparse networks as a $\ell_0$-regularized optimization problem (Louizos et al., 2017), resulting in a method that can be fully described in the optimization framework. To approximate $\ell_0$-regularization, we propose a smooth re-parameterization, allowing for the sub-network's structure to be directly learned with gradient-based methods. Unlike previous works, our re-parameterization is deterministic, proving more convenient for the tasks of pruning and ticket search, while also yielding faster training times.

Experimentally, our method offers superior performance when pruning VGG to extreme regimes, and is capable of finding winning tickets in Residual Networks trained on CIFAR-10 at a fraction of time taken by Iterative Magnitude Pruning. In particular, Continuous Sparsification successfully finds tickets in under 5 iterations, compared to 20 iterations required by Iterative Magnitude Pruning in the same setting. To further speed up the search for sub-networks, our method abdicates *parameter rewinding*, a key ingredient of Iterative Magnitude Pruning. By showing superior results without rewinding, our experiments offer insights on how ticket search should be performed.

## 2 RELATED WORK

### 2.1 LOTTERY TICKET HYPOTHESIS

The Lottery Ticket Hypothesis (Frankle & Carbin, 2019) states that for a network $f(x; w)$, $w \in \mathbb{R}^d$, and randomly-initialized parameters $w_0 \sim \mathcal{D}$, there exists a sparse sub-network, defined by a configuration $m \in \{0, 1\}^d$, $\|m\|_0 \ll d$, that, when trained from scratch, achieves higher performance than $f(x; w)$ while requiring fewer training iterations. The authors support this conjecture experimentally, showing that such sub-networks indeed exist: in particular, they can be discovered by repeatedly training, pruning, and re-initializing the network, through a procedure named Iterative Magnitude Pruning (IMP; Algorithm 1) (Frankle et al., 2019). More specifically, IMP alternates between: (1) training the weights $w$ of a network, (2) removing a fixed fraction of the weights with the smallest magnitude (pruning), and (3) *rewinding*: setting the remaining weights back to their original initialization $w_0$.

The sub-networks found by IMP, which indeed train faster and outperform their original, dense networks, are called *winning tickets*, and can generalize across datasets (Mehta, 2019; Soelen & Sheppard, 2019) and training methods (Morcos et al., 2019). In this sense, IMP can be a promising tool in applications that involve knowledge transfer, such as transfer or meta learning.

Zhou et al. (2019) perform extensive experiments to re-evaluate and better understand the Lottery Ticket Hypothesis. Relevant to this work is the fact that the authors propose a method to learn the binary mask $m$ in an end-to-end manner through SGD, instead of relying on magnitude-based pruning. The authors show that learning only the binary mask and not the weights is sufficient to achieve competitive performance, confirming that the learned masks are highly dependent on the initialized values $w_0$, and are also capable of encoding substantial information about a problem's solution.

### 2.2 SPARSE NETWORKS

The core aspect of searching for a winning ticket is finding a sparse sub-network that attains high performance relative to its dense counterpart. One way to achieve this is through pruning methods (LeCun et al., 1990), which follow a strategy to remove weights from a trained network while minimizing negative impacts on its performance. In Han et al. (2015), a network is iteratively trained and pruned using parameter magnitudes as criterion: this iterative, two-stage algorithm is shown to outperform "one-shot pruning": training and pruning the network only once.

Other methods attempt to approximate $\ell_0$ regularization on the weights of a network, yielding one-stage procedures that can be fully described in the optimization framework. In order to find a sparse

---

**Algorithm 1** Iterative Magnitude Pruning (Frankle et al., 2019)

---

1: Initialize $w \leftarrow w_0 \sim \mathcal{D}$ and $m \leftarrow \vec{1}^d$
2: Minimize $L(f(x; m \odot w))$ until $w_T$ is produced
3: Set $m_i = 0$ for the active weights with smallest magnitudes ($|w_{T,i}| \leq \tau$ and $m_i = 1$)
4: If satisfied, output ticket $f(x; m \odot w_k)$
5: Otherwise, set $w \leftarrow w_k$ and go back to step 2

---

**Algorithm 2** Iterative Stochastic Sparsification (inspired by Zhou et al. (2019))

---

1: Initialize $w \leftarrow w_0 \sim \mathcal{D}$, $s \leftarrow \vec{s}_0$
2: Minimize $\mathbb{E}_{m \sim \text{Ber}(\sigma(s))} [L(f(x; m \odot w))] + \lambda \|\sigma(s)\|_1$ until $w_T$ and $s_T$ are produced
3: If satisfied, output ticket $f(x; m \odot w_k)$, $m \sim \text{Ber}(\sigma(s_T))$
4: Otherwise, set $w \leftarrow w_k$, $s_i \leftarrow -\infty$ for $s_{i,T} < s_{i,0}$, and go back to step 2

---

setting $m \in \{0,1\}^d$ of a network $f(x; m \odot w)$, Srinivas et al. (2016) and Louizos et al. (2017) use a stochastic re-parameterization $m \sim \text{Bernoulli}(g(s))$ with $s \in \mathbb{R}^d$ and $g : \mathbb{R} \rightarrow [0,1]$ applied element-wise. First-order methods, coupled with gradient estimators, are then used to train both $w$ and $s$ to minimize the expected loss. This approach performs continuous parameter removal during training in an automatic fashion: any component $s_i$ of $s$ that assumes a value during training where $g(s_i) = 0$ effectively removes $w_i$ from the network. Moreover, approximating $\ell_0$ regularization has the advantage of not requiring a pruning strategy, which might be arbitrarily complex.

## 3 METHOD

Designing a method to quickly find winning tickets requires an efficient way to sparsify networks: ideally, sparsification should be done as early as possible in training, and the number of removed parameters should be maximized without harming the model's performance. In other words, sparsification must be *continuously* maximized following a trade-off with the performance of the network. This goal is not met by Iterative Magnitude Pruning: sparsification is done at discrete time steps, only after fully training the network, and optimal pruning rates likely depend on the model's performance and current sparsity: factors which are typically not accounted for – note that these are inherent characteristics of magnitude-based pruning.

In light of this, we turn to $\ell_0$-regularization methods for learning sparse networks, which consist of optimizing a clear trade-off between sparsity and performance. As we will see, performing sparsification continuously is not only straightforward, but done automatically by the optimizer.

### 3.1 CONTINUOUS SPARSIFICATION BY LEARNING DETERMINISTIC MASKS

We first frame the search for sparse networks as a loss minimization problem with $\ell_0$ regularization (Louizos et al., 2017; Srinivas et al., 2016):

$$\min_{w \in \mathbb{R}^d} L(f(x; w)) + \lambda \cdot \|w\|_0 \tag{1}$$

where $\lambda \geq 0$ controls the sparsity of the solution, and, with a slight abuse of notation, $L(f(x; w))$ denotes the loss incurred by the network $f(x; w)$ (*e.g.,* the cross-entropy loss over a training set). As $\ell_0$ regularization is typically intractable, we re-state the above minimization problem as:

$$\min_{\substack{w \in \mathbb{R}^d \\ m \in \{0,1\}^d}} L(f(x; m \odot w)) + \lambda \cdot \|m\|_1 \tag{2}$$

which uses the fact that, for $m \in \{0,1\}^d$, $\|m\|_0 = \|m\|_1$. The $\ell_1$ term can be minimized with subgradient descent, however the $m \in \{0,1\}^d$ constraint makes the above problem combinatorial and poorly suited for local search methods like SGD.

---

**Algorithm 3** Continuous Sparsification

1: Initialize $w \leftarrow w_0 \sim \mathcal{D}$, $s \leftarrow s_0$, $\beta \leftarrow \beta_0$
2: Minimize $L(f(x; \sigma(\beta \cdot s) \odot w)) + \lambda \left\| \sigma(\beta s) \right\|_1$ while increasing $\beta$, producing $w_T$, $s_T$, and $\beta_T$
3: If satisfied, output ticket $f(x; b(s_T) \odot w_k)$
4: Otherwise, set $s \leftarrow \min(\beta_T \cdot s_T, s_0)$, $\beta \leftarrow \beta_0$, (**optionally,** $w \leftarrow w_0$), and go back to step 2

---

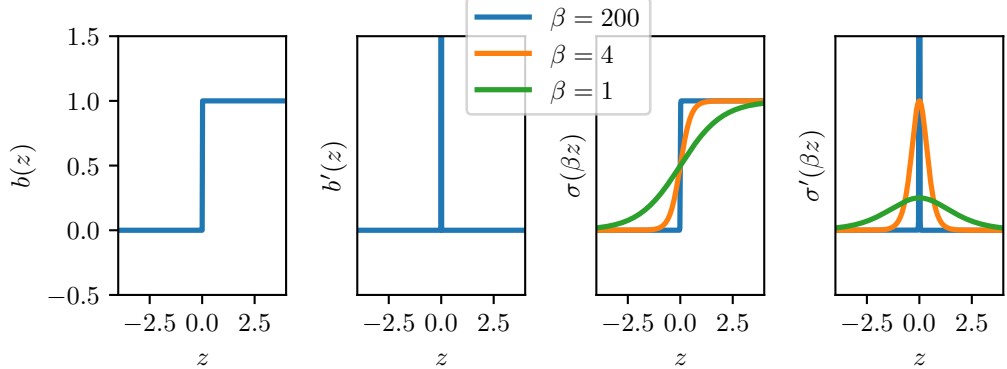

Figure 1: Illustration of our proposed re-parameterization $m = \sigma(\beta s)$, where $\sigma(z) = \frac{1}{1+e^{-z}}$ is the sigmoid function and $\beta$ acts as a temperature. As $\beta$ increases, $\sigma(\beta z)$ approaches $b(z)$, which can can be used to frame a $\ell_0$-regularized problem (Equation 4). Note that the gradients of $\sigma(\beta s)$ vanish as $\beta$ increases, suggesting that $\beta$ should be annealed slowly during training.

We can avoid the binary constraint $m \in \{0, 1\}^d$ by re-parameterizing $m$ as a function of a newly-introduced variable $s \in \mathbb{R}^d$. For example, Louizos et al. (2017) propose a stochastic mapping $s \mapsto m$ and use gradient methods to minimize the expected total loss, while using estimators for the gradients of $s$ (since $m$ is still binary). Having a stochastic mask (or, equivalently, a distribution over sub-networks) poses an immediate challenge for the task of finding tickets, as it is not clear which ticket should be chosen once a distribution over $m$ is learned. Moreover, relying on gradient estimators often causes gradients to have high variance, requiring longer training to reach optimality. Alternatively, we consider a deterministic parameterization $m = b(s)$, where $s \in \mathbb{R}^d_{\neq 0}$ and $b : \mathbb{R}_{\neq 0} \rightarrow \{0, 1\}$ is applied element-wise:

$$b(z) = \begin{cases} 1, & \text{if } z > 0 \\ 0, & \text{if } z < 0 \end{cases} \tag{3}$$

Applying this re-parameterization to Equation 2 yields:

$$\min_{\substack{w \in \mathbb{R}^d \\ s \in \mathbb{R}^d_{\neq 0}}} L(f(x; b(s) \odot w)) + \lambda \cdot \left\| b(s) \right\|_1 \tag{4}$$

Clearly, the above problem is again intractable, as it is still equivalent to the original $\ell_0$ problem in Equation 1. More specifically, the step function $b(z)$ is non-convex, and having zero gradients make gradient-based optimization ineffective. Instead, we consider the following smooth relaxation of $b(\cdot)$:

$$m := \sigma(\beta \cdot s) \tag{5}$$

where $\beta \in \mathbb{R}_{>0}$, and $\sigma$ is the sigmoid function $\sigma(z) = \frac{1}{1+e^{-z}}$, applied element-wise. By controlling $\beta$, which acts as a temperature parameter, we effectively interpolate between $\sigma(s)$, a smooth function well-suited for SGD, and $\lim_{\beta \to \infty} \sigma(\beta \cdot s) = b(z)$, our original goal, which brings computational hardness to the problem. Figure 1 illustrates this behavior. Note that, if $L(f(x; w))$ is continuous in $w$, then:

$$\min_{\substack{w \in \mathbb{R}^d \\ s \in \mathbb{R}^d_{\neq 0}}} \lim_{\beta \to \infty} L(f(x; \sigma(\beta s) \odot w)) + \lambda \cdot \|\sigma(\beta s)\|_1 = \min_{\substack{w \in \mathbb{R}^d \\ s \in \mathbb{R}^d_{\neq 0}}} L(f(x; b(s) \odot w)) + \lambda \cdot \|b(s)\|_1 \tag{6}$$

Although gradient methods will become ineffective as $\beta \to \infty$ due to vanishing gradients of $s$, we can increase $\beta$ while optimizing $s$ and $w$ with gradient descent. That is, our loss at each iteration will be a function of $\beta$ as follows:

$$L_\beta(w, s) = L(f(x; \sigma(\beta s) \odot w)) + \lambda \cdot \|\sigma(\beta \cdot s)\|_1 \tag{7}$$

How does the soft mask $m = \sigma(\beta \cdot s)$ behave as we minimize $L_\beta(w, s)$ while increasing $\beta$? As $\beta \to \infty$, every negative component of $s$ will be mapped to 0, effectively removing its correspondent weight parameter from the network. While analytically the weights will never truly be zeroed-out, limited numerical precision has the fortunate side-effect of causing actual sparsification to the network during training, as long as $\beta$ is increased to a large enough value.

In a nutshell, we learn sparse networks by minimizing $L_\beta(w, s)$ for $T$ parameter updates with gradient descent while jointly annealing $\beta$: producing $w_T$, $s_T$ and $\beta_T$, which is ideally large enough such that, *numerically* [1], $\sigma(\beta_T \cdot s_T) = b(s_T)$. In case $m$ is truly required to be binary (as in the task of finding tickets), the dependence on numerical imprecision can be avoided by directly outputting $m = b(s_T)$ at the end of training.

Finally, note that minimizing $L_\beta$ while increasing $\beta$ is *not* generally equivalent to minimizing the original $\ell_0$-regularized problem. Informally, the former aims to solve $\lim_{\beta \to \infty} \min_{w,s} L_\beta(w, s)$, while the $\ell_0$ problem is $\min_{w,s} \lim_{\beta \to \infty} L_\beta(w, s)$.

## 3.2 Ticket Search through Continuous Sparsification

The method presented above offers a direct alternative to magnitude-based pruning when performing ticket search, but a few considerations must follow. Most importantly, when searching for winning tickets, there is a strict constraint that the learned mask $m$ be binary: otherwise, one can also learn the magnitude of the weights, defeating the purpose of finding sub-networks that can be trained *from scratch*. To guarantee that the output mask satisfies this constraint regardless of numerical precision, we always output $b(s_T)$ instead of $\sigma(\beta_T \cdot s_T)$.

Additionally, we also incorporate two techniques from successful methods for learning sparse networks and searching for winning tickets. First, motivated by Han et al. (2015), where it is shown that iteratively pruning a network yields improved sparsity compared to pruning it only once, we enable "kept" weights – those whose corresponding component of $s$ is positive after many iterations – to be removed from the network at a later stage. More specifically, when $\beta$ becomes large after $T$ gradient descent updates, the gradients of $s$ vanish and weights will no longer be removed from the network. To avoid this, we set $s \leftarrow \min(\beta_T \cdot s_T, s_0)$, effectively resetting the soft mask parameters for the remaining weights while at the same time not interfering with weights that have been removed. This is followed by a reset on the temperature, $\beta \leftarrow \beta_0$, to allow training of $s$ once again.

Second, we perform parameter rewinding, following Frankle & Carbin (2019), which is a key component of Iterative Magnitude Pruning. More specifically, after $T$ gradient descent steps, we reset the weight values back to an earlier stage $w \leftarrow w_k$, where $k \ll T$. Even though experimental results in Frankle & Carbin (2019) suggest that rewinding is necessary for successful ticket search, we leave rewinding as an optinal component of our algorithm: as we will see empirically, it turns out that ticket search is possible without rewinding weights. Our proposed algorithm to find winning tickets is presented as Algorithm 3, and referred simply as "Continuous Sparsification".

---

[1]We observed in our experiments that a final temperature of 500 is sufficient for iterates of $s$ when training with SGD with 32-bit precision. The required temperature is likely to depend on the how $s$ is numerically represented, as in reality our method relies on numerical imprecision.

# 4 EXPERIMENTS

Our experiments aim at comparing different methods on the task of finding winning tickets in neural networks, hence our evaluation focuses on the generalization performance of each ticket (sub-network) *when trained from scratch* (or from an iterate in early-training). Additionally, we measure the cost of the search procedure: the number of training epochs to find tickets with varying performance and sparsity.

Besides comparing our proposed method to Iterative Magnitude Pruning (Algorithm 1), we also design a baseline method, Iterative Stochastic Sparsification (ISS, Algorithm 2), motivated by the procedure in Zhou et al. (2019) to find a binary mask $m$ with gradient descent in an end-to-end fashion. More specifically, ISS uses a stochastic re-parameterization $m \sim \text{Bernoulli}(\sigma(s))$ with $s \in \mathbb{R}^d$, and trains $w$ and $s$ jointly with gradient descent and the straight-through estimator (Bengio et al., 2013). When ran for multiple iterations, all components of the mask parameters $s$ which have decreased in value from initialization are set to $-\infty$, such that the corresponding weight is permanently removed from the network. While this might look arbitrary, we observed empirically that ISS was unable to remove weights quickly without this step unless $\lambda$ was chosen to be large – in which case the model's performance decrease in exchange for sparsity. The hyperparameters used in this section were chosen based on analysis presented in Appendix (...), where we study how the pruning rate affects IMP, and how $\lambda$, $s_0$ and $\beta_T$ interact in CS.

## 4.1 CONVOLUTIONAL NEURAL NETWORKS

We train a neural network with 6 convolutional layers on the CIFAR-10 dataset (Krizhevsky, 2009), following Frankle & Carbin (2019). The network consists of 3 blocks of 2 resolution-preserving convolutional layers followed by $2 \times 2$ max-pooling, where convolutions in each block have $64, 128$ and $256$ channels, a $3 \times 3$ kernel, and are immediately followed by ReLU activations. The blocks are followed by fully-connected layers with $256, 256$ and $10$ neurons, with ReLUs in between. The network is trained with Adam (Kingma & Ba, 2015) with a learning rate of $0.0003$ and a batch size of $60$.

**Learning a Supermask:** As a first baseline, we consider the task of learning a "supermask" (Zhou et al., 2019): a binary mask $m$ that, when applied to a network with randomly initialized weights, yields performance competitive to that of training its weights. This task is equivalent to pruning a randomly-initialized network, or learning an architecture that performs well prior to training with a fixed initialization. We compare ISS and CS , where each method is run for a single iteration composed of 100 epochs. When ran for a single iteration, ISS is equivalent to the algorithm proposed in Zhou et al. (2019) to learn a supermask, referred here as simply Stochastic Sparsification. We control the sparsity of the learned masks by varying $s_0$ between $-5$ and $5$ for Stochastic Sparsification (which showed to be more effective than varying $\lambda$), while for Continuous Sparsification we vary $\lambda$ between $10^{-11}$ and $10^{-7}$ (which results in stable and consistent training, unlike varying $s_0$). SS uses SGD with a learning rate of 100 to learn its mask parameters, while CS uses Adam with $3 \times 10^{-4}$.

Results are presented in Figure 2: CS outperforms SS in terms of both training speed and the quality of the learned mask. In particular, CS finds masks with over $75\%$ sparsity that yield over $75\%$ test accuracy, while the performance of masks found by SS decrease when sparsity is over $50\%$. Moreover, CS makes faster progress in training, showing that optimizing a deterministic mask is indeed faster than learning a distribution over masks through stochastic re-parameterizations.

**Finding Winning Tickets:** We run IMP and ISS for a total of 30 iterations, each consisting of 40 epochs. Parameters are trained with Adam (Kingma & Ba, 2015) with a learning rate of $3 \times 10^{-4}$, following Frankle & Carbin (2019). For IMP, we use pruning rates of $15\%/20\%$ for convolutional/dense layers. We initialize the Bernoulli parameters of ISS with $s_0 = \vec{1}$, and train them with SGD and a learning rate of 20, along with a $\ell_1$ regularization of $\lambda = 10^{-8}$. For CS , we anneal the temperature from $\beta_0 = 1$ to $\beta_0 = 250$ following an exponential schedule ($\beta_t = 250^{\frac{t}{T}}$), training both the weights and the mask with Adam and a learning rate of $3 \times 10^{-4}$.

To test whether our method is capable of finding winning tickets in a limited amount of time, we limit each run of CS to 4 iterations only, in contrast with IMP and ISS which are run for 30. We perform 6 runs of CS , each with a different value for the mask initialization $s_0$: $-0.05, -0.03, -0.02, -0.01$, $-0.005, 0$, keeping $\lambda = 10^{-10}$, such that sparsification is not enforced during training, but heavily

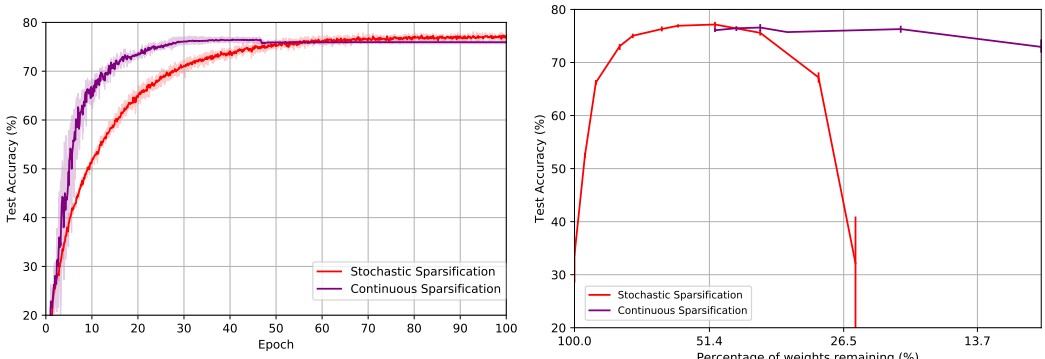

Figure 2: Learning a binary mask with weights frozen at initialization with Stochastic Sparsification (SS, Algorithm 2 with one iteration) and Continuous Sparsification (CS), on a 6-layer CNN on CIFAR-10. **Left:** Training curves with hyperparameters for which masks learned by SS and CS were both approximately $50\%$ sparse. CS learns the mask significantly faster while attaining similar early-stop performance. **Right:** Sparsity and test accuracy of masks learned with different settings for SS and CS: our method learns sparser masks while maintaining test performance, while SS is unable to successfully learn masks with over $50\%$ sparsity.

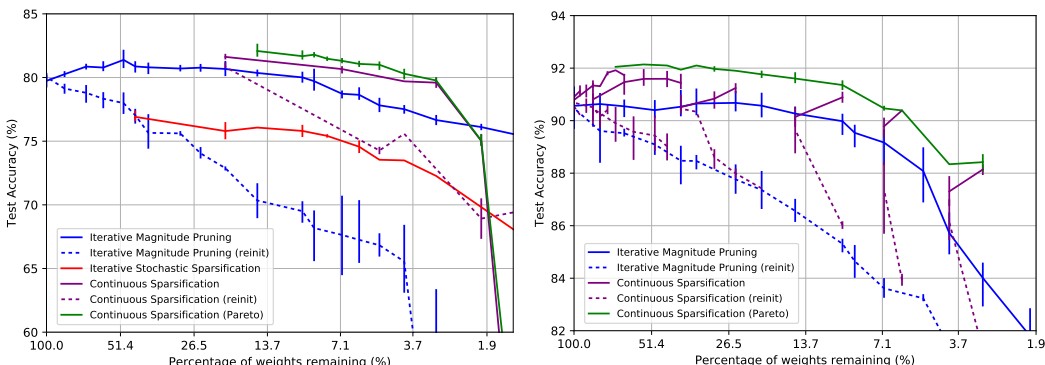

Figure 3: Test accuracy of tickets found by different methods on CIFAR-10. Error bars depict variance across 3 runs. **Left:** Performance of tickets found on a 6-layer CNN, when trained from scratch. **Right:** Performance of tickets found on a ResNet 20, when rewinded to the second training epoch. In both experiments, tickets found by CS outperform ones found by IMP. In most cases, CS successfully finds winning tickets in 2 iterations (purple curves).

biased at initialization. In order to evaluate how consistent our method is, we repeat each run with 3 different random seeds so that error bars can be computed.

Figure 3 (**left**) presents the quality of tickets found by each method, measured by their test accuracy when trained from scratch. To illustrate the quality of the tickets that can be found by Continuous Sparsification, we plot the Pareto curve (green) of the tickets founds with the 6 different values for $s_0$. With $s_0 = -0.03$, in only 2 iterations CS finds a ticket with over $77\%$ sparsity (first marker of purple curve) which outperforms *every* ticket found by IMP in its 30 iterations. The Pareto curve of CS strictly dominates IMP for tickets with more less than $97\%$ sparsity, where ticket performance is superior or similar to the original dense network.

In terms of computational time, the total cost to run CS with the 6 different values for $s_0$ is lower than performing a single run of IMP for 30 iterations, even though CS takes $15\%$ extra time per epoch due to the mask parameters. This shows the potential of our model even in the setting where a specific sparsity is desired for the tickets. When run in parallel, CS takes less wall-clock time to find all tickets in the Pareto curve than to run IMP for 5 iterations.

### 4.2 FINDING WINNING TICKETS IN RESIDUAL NETWORKS WITHOUT REWINDING

Searching for tickets in realistic models is not as straightforward as finding tickets in a small CNN, and might require new strategies. Frankle et al. (2019) show that IMP fails at finding winning tickets in ResNets (He et al., 2016) unless the learning rate is smaller than the recommended value, leading to worse overall performance and defeating the purpose of ticket search. However, the authors propose a slight modification to IMP that enables search for winning tickets to be successful on complex networks: instead of training from scratch, tickets are initialized with weights from early training.

With this in mind, we evaluate how Continuous Sparsification performs in the time-consuming task of finding winning tickets in a ResNet-20 [2] (He et al., 2016) trained on CIFAR-10: a setting where IMP might take over 10 iterations (850 epochs) to succeed. We follow the setup in Frankle & Carbin (2019) and Frankle et al. (2019): in each iteration, the network is trained with SGD, a learning rate of $0.1$, and a momentum of $0.9$ for a total of $85$ epochs, using a batch size of $128$. The learning rate is decayed by a factor of $10$ at epochs $56$ and $71$, and a weight decay of $0.0001$ is applied to the weights (for CS , we do *not* apply weight decay to the mask parameters $s$). The two skip-connections that perform $1 \times 1$ convolutions and the output layer are not removable: for IMP, their parameters are not pruned, while for CS their weights do not have a correspondent mask $m$ nor mask parameters $s$.

When training the returned tickets in order to evaluate their performance, we initialize their weights with the iterates from the end of epoch 2 (780 parameter updates), similarly to Frankle et al. (2019). Unlike when searching for winning tickets in the 6-layer CNN, IMP performs global pruning, removing $20\%$ of the remaining parameters with smallest magnitude, ranked globally (across different layers). IMP runs for a total of $30$ iterations, while CS is limited to only $5$ iterations for each run. The sparsity of the tickets found by CS is controlled by varying the mask initialization $s_0 \in \{-0.3, -0.2, -0.1, -0.05, -0.03, 0, 0.03, 0.05, 0.1, 0.2, 0.3\}$ (a total of 11 values). To allow for even faster ticket search, we run CS **without parameter rewinding**: that is, the weights $w$ are transferred from one iteration to another, removing the need to re-train the network as the method progresses through iterations. For both CS and IMP, each run is repeated with 3 different random seeds.

The results presented in Figure 3 (**right**) show that CS is able to successfully find winning tickets with varying sparsity in under 5 iterations. Once again, the Pareto curve strictly dominates IMP, and variance across runs is smaller than IMP's. Most notably, CS is capable of quickly sparsifying the network in a single iteration (first marker of each purple curve), and typically finds better tickets than IMP after only 2 rounds (compare blue curve and second marker of each purple curve), regardless of sparsity. When run in parallel, 2 iterations suffice for CS to find tickets that outperform the ones found by IMP.

We observed that not performing rewinding caused the performance of tickets with high sparsity to quickly degrade after 2 or more iterations of CS. We speculate that, when rewinding is not performed between iterations, the distance between $w_k$ and the parameter iterates produced by gradient descent $w_t$ increase significantly with the number of iterations. This in turn can result in the learned mask $m_T$ to be highly sub-optimal for weight values $w_k$ ($k \ll T$) which are used to re-train the ticket. This suggests that in order to avoid re-training the network and hence make the search for winning tickets more efficient, rewinding should not be performed between iterations. In this case, the search must complete quickly, before performance degradation occurs due to "overtraining", requiring optimal ways to perform sparsification without negatively impacting the model's performance.

### 4.3 PRUNING VGG

Our experiments show that Continuous Sparsification is capable of finding tickets quickly and consistently, and we attribute its success to its deterministic re-parameterization of the binary mask. Here, we evaluate our method a pruning technique, to better assess whether our proposed re-parameterization is advantageous only in terms of training time, or also in respect to the quality of the learned masks.

For this task, we train a VGG (Simonyan & Zisserman, 2015) on the CIFAR-10 dataset, following the protocol in Frankle & Carbin (2019): the network is trained with SGD and an initial learning rate of $0.1$, which is decayed by a factor of $10$ at epochs $80$ and $120$. After $160$ training epochs, the network

---

[2]We used the same network as Frankle & Carbin (2019) and Frankle et al. (2019), where it is referred as a ResNet 18.

is sparsified and then fine-tuned for 40 epochs with a learning rate of $0.001$. We evaluate previously described methods when executed for a single iteration (one-shot pruning): Continuous Sparsification, Magnitude Pruning (IMP with 1 iteration) (Han et al., 2015), and Stochastic Sparsification (ISS with 1 iteration), which is similar to methods in Zhou et al. (2019), Srinivas et al. (2016), and Louizos et al. (2017).

At the sparsification step, IMP performs global pruning, ISS fixes the binary mask $m$ to be the maximum likelihood one under $\text{Ber}(\sigma(s))$ (which performed better than sampling from the distribution), and CS changes the parameterization of the mask from $\sigma(\beta s)$ to $b(s)$ (or, equivalently, weights $w_i$ where $s_i < 0$ are removed). We use a momentum of $0.9$, a weight decay of $0.0001$ (not applied to $s$), and a batch-size of $64$. Following Frankle & Carbin (2019), sparsification is not applied to batch normalization nor the final linear layer.

To evaluate each method when finding masks with different sparsity levels, we run IMP with global pruning rates $50\%$, $75\%$, $80\%$, $85\%$, $90\%$, $95\%$, $97.5\%$, $98\%$, $98.5\%$, $99\%$, $99.5\%$, $99.75\%$, and ISS and CS with initial mask values $-0.3$, $-0.25$, $-0.2$, $-0.15$, $-0.1$, $-0.05$, $-0.01$, $-0.005$, $-0.001$, $0$. Results are shown in Figure 4: both magnitude pruning and stochastic $\ell_0$ regularization (Stochastic Sparsifi-

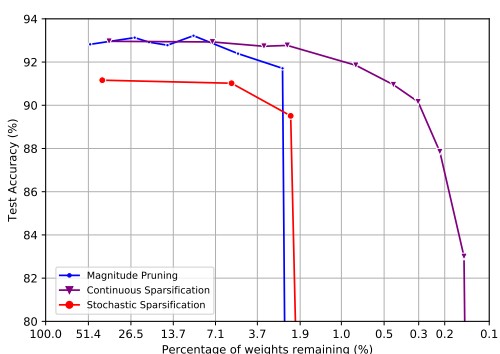

Figure 4: Performance of different methods when performing one-shot pruning on VGG. CS maintains over $90\%$ test accuracy after removing $99.7\%$ of the weights, while other methods fail to successfully remove more than $98\%$ of the parameters.

cation) fail at removing over $98\%$ of the weights without severely degrading the performance of the model. On the other hand, Continuous Sparsification successfully removes $99.7\%$ of the parameters in the convolutional layers while still yielding over $90\%$ test accuracy. When taken to the extreme, our method is capable of removing $99.85\%$ of the weights and still yield over $83\%$ accuracy.

The dramatic performance difference between stochastic and continuous sparsification shows that our proposed deterministic re-parameterization is key to achieve superior results in both network pruning and ticket search. The fact that it outperforms magnitude pruning, a standard technique in the pruning literature, suggests that further exploration of $\ell_0$-based methods could yield significant advances in pruning techniques.

## 5 DISCUSSION

With Frankle & Carbin (2019), we now realize that sparse sub-networks can indeed be successfully trained from scratch, putting in question the belief that overparameterization is required for proper optimization of neural networks. Such sub-networks, called winning tickets, can be potentially used to significantly decrease the required resources for training deep networks, as they are shown to transfer between different, but similar, tasks (Mehta, 2019; Soelen & Sheppard, 2019).

Currently, the *search* for winning tickets is a poorly explored problem, where Iterative Magnitude Pruning (Frankle & Carbin, 2019) stands as the only algorithm suited for this task, and it is unclear whether its key ingredients – post-training magnitude pruning and parameter rewinding – are the correct choices for the task. Here, we approach the problem of finding sparse sub-networks as an $\ell_0$-regularized optimization problem, which we approximate through a smooth, parameterized relaxation of the step function. Our proposed algorithm for finding winning tickets, Continuous Sparsification, removes parameters automatically and continuously during training, and can be fully described by the optimization framework. We show empirically that, indeed, post-training pruning might not be a sensible choice for finding winning tickets, raising questions on how the search for tickets differs from standard network compression. With this work, we hope to further motivate the problem of *quickly* finding tickets in overparameterized networks, as recent work suggests that the task might be highly relevant to transfer learning and mobile applications.

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

APPENDIX

# A  HYPERPARAMETER ANALYSIS

## A.1  CONTINUOUS SPARSIFICATION

In this section, we study how the hyperparameters of Continuous Sparsification affect its performance in terms of sparsity and performance of the found tickets. More specifically, we consider the following hyperparameters:

- Final temperature $\beta_T$: the final value for $\beta$, which controls how smooth the parameterization $m = \sigma(\beta s)$ is.
- $\ell_1$ penalty $\lambda$: the strength of the $\ell_1$ regularization applied to the soft mask $\sigma(\beta s)$, which promotes sparsity.
- Mask initial value $s_0$: the value used to initialize all components of the soft mask $m = \sigma(\beta s)$, where smaller values promote sparsity.

Our setup is as follows: to analyze how each of the 3 hyperparameters impact the performance of Continuous Sparsification, we train a ResNet 20 on CIFAR-10 (following the same protocol from Section 4.2), varying one hyperparameter while keeping the other two fixed. To capture how hyperparameters interact with each other, we repeat the described experiment with different settings for the fixed hyperparameters.

Since different hyperparameter settings naturally yield vastly distinct sparsity and performance for the found tickets, we report relative changes in accuracy and in sparsity.

In Figure 5, we vary $\lambda$ between 0 and $10^{-8}$ for three different $(s_0, \beta_T)$ settings: $(s_0 = -0.2, \beta_T = 100)$, $(s_0 = 0.05, \beta_t = 200)$, and $(s_0 = -0.3, \beta_T = 100)$. As we can see, there is little impact on either the performance or the sparsity of the found ticket, except for the case where $s_0 = 0.05$ and $\beta_T = 200$, for which $\lambda = 10^{-8}$ yields slightly increased sparsity.

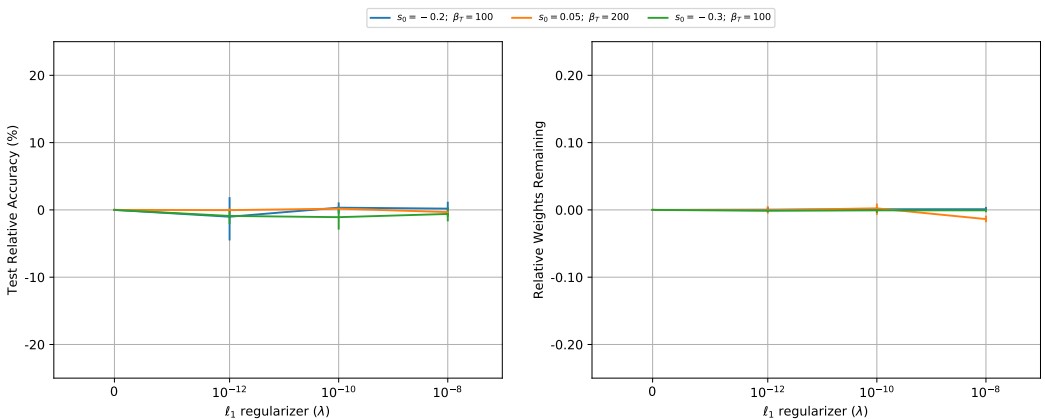

Figure 5: Impact on relative test accuracy and sparsity of tickets found in a ResNet 20 trained on CIFAR-10, for different values of $\lambda$ and fixed settings for $\beta_T$ and $s_0$.

Next, we consider the fixed settings $(s_0 = -0.2, \lambda = 10^{-10})$, $(s_0 = 0.05, \lambda = 10^{-12})$, $(s_0 = -0.3, \lambda = 10^{-8})$, and proceed to vary the final temperature $\beta_T$ between 50 and 200. Figure 6 shows the results: in all cases, a larger temperature of 200 yielded better accuracy. However, it decreased sparsity compared to smaller temperature values for the settings $(s_0 = -0.2, \lambda = 10^{-10})$ and $(s_0 = -0.3, \lambda = 10^{-8})$, while at the same time increasing sparsity for $(s_0 = 0.05, \lambda = 10^{-12})$. While larger temperatures appear beneficial and might suggest that even higher values should be used, note that, the larger $\beta_T$ is, the earlier in training the gradients of $s$ will vanish, at which point training of the mask will stop. Since the performance for temperatures between 100 and 200 does not change significantly, we recommend values around 150 or 200 when either pruning or performing ticket search.

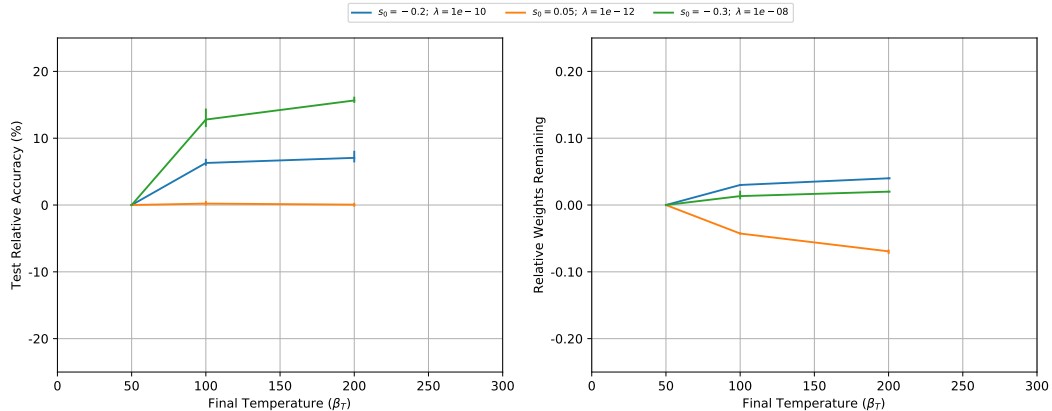

Figure 6: Impact on relative test accuracy and sparsity of tickets found in a ResNet 20 trained on CIFAR-10, for different values of $\beta_T$ and fixed settings for $\lambda$ and $s_0$.

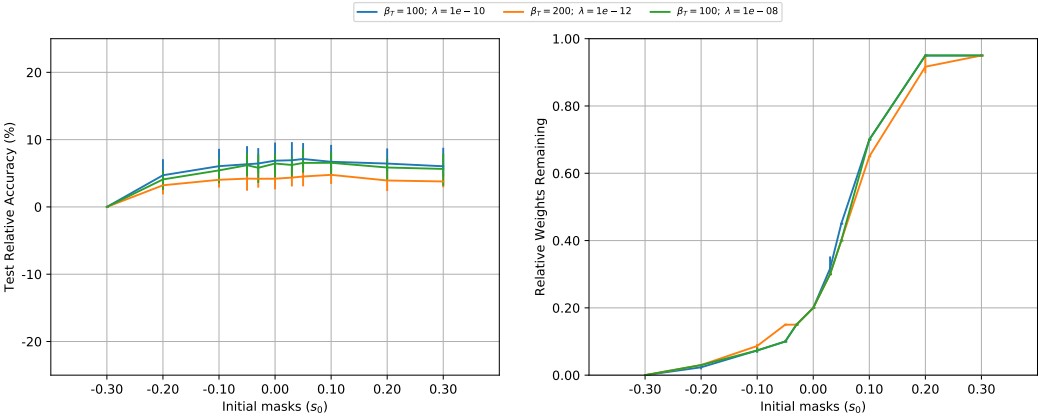

Figure 7: Impact on relative test accuracy and sparsity of tickets found in a ResNet 20 trained on CIFAR-10, for different values of $s_0$ and fixed settings for $\beta_T$ and $\lambda$.

Lastly, we vary the initial mask value $s_0$ between $-0.3$ and $+0.3$, with hyperpameter settings $(\beta_T = 100, \lambda = 10^{-10})$, $(\beta_T = 200, \lambda = 10^{-12})$, and $(\beta_T = 100, \lambda = 10^{-8})$. Results are given in Figure 7: unlike the exploration on $\lambda$ and $\beta_T$, we can see that $s_0$ has a strong and consistent effect on the sparsity of the found tickets. For this reason, we suggest proper tuning of $s_0$ when the goal is to achieve a specific sparsity value. Since the percentage of remaining weights is monotonically increasing with $s_0$, we can perform binary search over values for $s_0$ to achieve any desired sparsity level. In terms of performance, lower values for $s_0$ naturally lead to performance degradation, since sparsity quickly increases as $s_0$ becomes more negative.

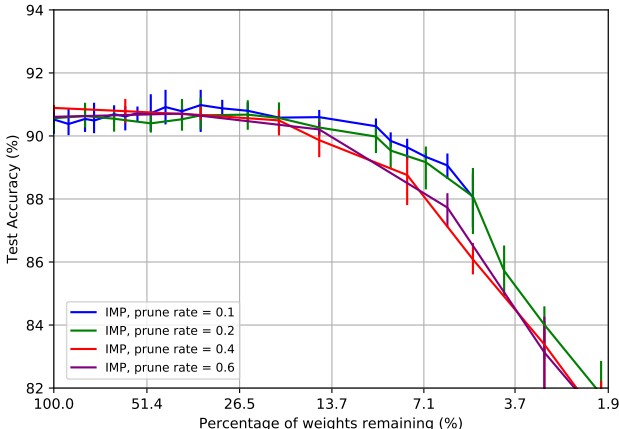

Figure 8: Performance of tickets found by Iterative Magnitude Pruning in a ResNet 20 trained on CIFAR, for different pruning rates.

## A.2 ITERATIVE MAGNITUDE PRUNING

Here, we assess whether the running time of Iterative Magnitude Pruning can be improved by increasing the amount of parameters pruned at each iteration. The goal of this experiment is to evaluate if Continuous Sparsification offers faster ticket search only because it prunes the network more aggressively than IMP, or because it is truly more effective in how parameters are chosen to be removed.

Following the same setup as the previous section, we train a ResNet 20 on CIFAR-10. We run IMP for 30 iterations, performing global pruning with different pruning rates at the end of each iteration. Figure 8 shows that the performance of tickets found by IMP decays when the pruning rate is increased to $40\%$. In particular, the final performance of found tickets is mostly monotonically decreasing with the number of remaining parameters, suggesting that, in order to find tickets which outperform the original network, IMP is not compatible with more aggressive pruning rates.

