# OpenReview forum: "Winning the Lottery with Continuous Sparsification"
_ICLR.cc/2020/Conference — Reject_

### Official Review · AnonReviewer1 · 2019-10-22
**Official Blind Review #1**

**Rating:** 6

**Review:**

This paper proposes a novel objective function that can be used to jointly optimize a classification objective while at the same time encourage sparsification in a network. The lottery ticket hypothesis and associated work shows that the iterative pruning of a network can lead to a sparse network that performs with high accuracy. On the other hand, the work of Zhou et al. shows that sparse masks (dubbed "supermasks") may be learned without training the parameters of the network. In a sense, this paper tries to combine these ideas by simultaneously training a network while also optimizing the mask.

I think this paper serves as a reasonable contribution to the ever-growing "lottery ticket hypothesis" body of work. The paper is mostly clear, and the idea for joint optimization is very reasonable. It's not tremendously original (in that it basically combines two ideas that are already in the literature), but in spite of that, I still think this paper warrants being accepted to ICLR.

For me, the most interesting scientific point is about the issue of rewinding. In particular, the fact that continuous sparsification can find winning tickets without any parameter rewinding is fascinating and deserves further investigation. Do the authors have any sense for why this works, when prior work suggests that rewinding is necessary for sufficiently complicated models and datasets?

A minor point, I think there's a typo on page 6 in that the paragraph beginning "Results are presented in Figure 2" both instances of "SP" should be "SS" instead.

**Experience Assessment:**

I have read many papers in this area.

**Review Assessment: Checking Correctness Of Derivations And Theory:**

I carefully checked the derivations and theory.

**Review Assessment: Checking Correctness Of Experiments:**

I assessed the sensibility of the experiments.

**Review Assessment: Thoroughness In Paper Reading:**

I read the paper at least twice and used my best judgement in assessing the paper.

---

> ### Author Response · Authors · 2019-11-13
> **Response to reviewer 1**
>
> Thank you for your comments. We address your points individually below — please let us know if we can clarify or address any further concerns.
>
>
>
> - “it basically combines two ideas that are already in the literature”:
>
> Our method significantly differs from both Zhou et al. and Louizos et al. as we use a novel deterministic re-parameterization for the mask, which avoids the need of gradient estimators and makes training and thus ticket search significantly faster.
>
> The baseline ‘Iterative Stochastic Sparsification’ (Algorithm 2) is what would be a naive combination of ticket search and the method in Zhou et al. that directly optimizes the mask. Our experiments show that our method, Continuous Sparsification, yields significantly better results in both learning a supermask (Figure 2) and ticket search (compare green and red curves in the left plot of Figure 3). In particular, the baseline Iterative Stochastic Sparsification underperforms Iterative Magnitude Pruning, showing that our deterministic re-parameterization was indeed necessary to push the state-of-the-art performance in ticket search.
>
> To further support this, we have added new results showing that, when pruning a VGG network trained on CIFAR, our method outperforms both magnitude pruning and stochastic l0 regularization. Results are presented in Section 4.3: our method successfully prunes 99.7% of the parameters while still maintaining over 90% test accuracy, while both magnitude pruning and stochastic sparsification suffer from severe performance degradation when over the pruning rate is over 98%, achieving less than 70% accuracy. These results show that our deterministic re-parameterization is fundamentally different than the stochastic re-parameterizations proposed in previous works such as  Zhou et al.: it provides superior performance in both ticket search and one-shot pruning, while at the same time being simpler by not requiring gradient estimators.
>
> ———————————————-
>
> - “In particular, the fact that continuous sparsification can find winning tickets without any parameter rewinding is fascinating and deserves further investigation. Do the authors have any sense for why this works, when prior work suggests that rewinding is necessary for sufficiently complicated models and datasets?”
>
> We hypothesize that rewinding is necessary only if ticket search consists of many training epochs: in this case, the parameters can ‘move’ far from the initialization values, at which point the mask might not be suitable for the parameters close initialization. More specifically, after T parameter updates, we have weights w_T and mask m_T, where m_T was computed from either w_T or w_{T-1} (using the magnitudes of w_T in IMP, or after a gradient update from {T-1} in CS). However, since the ticket is given by (w_k, m_T), for small k, if w_T differs too much from w_k (say, ||w_T - w_k|| is large), then m_T might be highly suboptimal for w_k, and the ticket can fail to be successfully re-trained. Since in our method the number of updates T is significantly smaller than in IMP, the need to rewind the weights back to w_k is diminished. This is briefly described in the last paragraph of Section 4.2.
>
> ———————————————-
>
> “"Results are presented in Figure 2" both instances of "SP" should be "SS" instead.”
>
> Thanks for pointing out the typo — we have fixed it in the revision.

---

### Official Review · AnonReviewer2 · 2019-10-24
**Official Blind Review #2**

**Rating:** 3

**Review:**

This work propose a new iterative pruning methods named Continuous Sparsification. It will continuously prune the current weight until it reaches the target ratio instead of iterative prune the weight to specific ratio. The author gives a good analysis but the experiment is not yet convincing enough.

1) This work actually presents compression algorithm with little connection with lottery ticket. As a lottery ticket discussion, it does not give the comparison between lottery ticket and random initialization based on new pruning method. As a pruning method, it does not show the results on common models like VGG and DenseNet with different depth. Figure. 3 only gives results of ResNet-18 while the setting is not the best setting. Normally we need to train at least 120 epochs. It also does not give the experiment on ImageNet. Thus making the conclusion less meaningful

2) The author should also compare the continuous sparsification with one-shot pruning methods (non-iterative) to see the advantage of continuous sprsification.

**Experience Assessment:**

I have published one or two papers in this area.

**Review Assessment: Checking Correctness Of Derivations And Theory:**

I carefully checked the derivations and theory.

**Review Assessment: Checking Correctness Of Experiments:**

I carefully checked the experiments.

**Review Assessment: Thoroughness In Paper Reading:**

I read the paper at least twice and used my best judgement in assessing the paper.

---

> ### Author Response · Authors · 2019-11-13
> **Response to reviewer 2**
>
> Thank you for the review and comments. We address your points individually below — please let us know if we can clarify or address any further concerns.
>
>
>
> - “little connection to the lottery ticket”
>
> Our method was designed in the scope of finding winning tickets in large networks. A notable difference between pruning and ticket search (as done with IMP) is that ticket search requires more iterations compared to pruning, resulting in computational costs that might be prohibitive (which is not typically the case for pruning). The core idea of Continuous Sparsification is to use a deterministic re-parameterization to learn the masks, hence avoiding having to minimize a stochastic objective or use gradient estimators which create additional variance. The main design goal of Continuous Sparsification is to be fast: a concern that is due to ticket search.
>
> ———————————————-
>
> - “it does not give the comparison between lottery ticket and random initialization based on new pruning method.”
>
> We have added curves to Figure 3 (dashed lines) presenting the performance of sub-networks when they are randomly re-initialized instead. The performance is visibly inferior to sub-networks whose parameters have been rolled back to their original initialization, which agrees with the observations in Frankle & Carbin.
>
> ———————————————-
>
> - “As a pruning method, it does not show the results on common models like VGG and DenseNet“
>
> We have added new results showing that, when pruning a VGG trained on CIFAR, our method outperforms both magnitude pruning and stochastic l0 regularization by a wide margin. Results are presented in Section 4.3: our method successfully prunes 99.7% of the parameters while still maintaining over 90% test accuracy, while both magnitude pruning and stochastic sparsification suffer from severe performance degradation when over the pruning rate is over 98%, achieving less than 70% accuracy.
>
> ———————————————-
>
> - “gives results of ResNet-18 while the setting is not the best setting. Normally we need to train at least 120 epochs”
>
> We have precisely followed the training protocol in Frankle et al., where a ResNet is trained for over 15 iterations, each consisting of 85 epochs. We have also used exactly the same hyperparameters, including learning rate, batch size, weight decay, and learning rate schedule. This was done for multiple reasons, including to have a fair comparison between the two methods, and to show readers that, as our results with Iterative Magnitude Pruning match the ones reported in Frankle et al., our implementation is consistent with theirs.
>
> ———————————————-
>
> - “It also does not give the experiment on ImageNet”
>
> We have not performed these experiments due to the computational costs of fully training an ImageNet model for many iterations in a sequential fashion. In particular, Frankle et al. (“Stabilizing the Lottery Ticket Hypothesis”) train a ResNet 50 for over 1300 epochs on ImageNet. Nonetheless, we are currently working on evaluating our method when training a ResNet 50 on ImageNet, and will add them to the camera-ready version of the paper.

---

### Official Review · AnonReviewer4 · 2019-11-03
**Official Blind Review #4**

**Rating:** 3

**Review:**


To the authors of paper 2504: I have posted a private comment for the reviewer/AC discussion period based on your author responses and your revised paper. I want you to be able to see my full response, but I can't post additional public comments to the paper. As such, I'm editing my review with exactly what I sent to the AC.

Thank you to the authors for your thoughtful rebuttal and for updating the paper with both new text and (more impressively) new experiments. I read all three of your rebuttals and re-read the revised paper in detail. My comment to the AC is at the bottom of this message. I don't know if you will get an alert that the review was updated, but I hope you get a chance to take a look.

===========================================

EXECUTIVE SUMMARY OF REVIEW

Summary of paper: This paper proposes a technique that simultaneously trains a neural network and learns a pruning mask. The goal of this technique is to make it faster to retroactively find sparse subnetworks that, from a point early in training, could have trained in isolation to the same performance as the full network ("winning tickets"). The best subnetworks found by this technique outperform those found by existing techniques [1] at every sparsity level.

Summary of review: The technique introduces several new hyperparameters whose values are asserted without describing the extent of the search necessary to find them; it unclear whether the cost of the search cancels out the efficiency gains. In addition, the proposed technique is inconsistent across runs. The sparsity and accuracy of the subnetworks it produces vary greatly, and there are no hyperparameters to explicitly control these outcomes. It is also unclear whether results vary from run to run even with the same hyperparameters.  As such, this technique is not clearly a cost reduction as compared to existing approaches in [1] when it comes to studying lottery tickets. Moreover, the paper implicitly proposes a new pruning technique, and it should be evaluated against other related techniques in the pruning literature. Finally, the evaluation needs more rigor.

Conclusion: It is difficult to determine whether the technique is an improvement over existing methods since the true costs of using it in practical workflows are unclear. Weak reject.

Opportunity to improve score: Include a more detailed analysis of the overall costs of finding winning tickets at a target sparsity using the proposed technique, particularly including the costs of hyperparameter search necessary to do so. Include multiple replicates of experiments, experiments on more networks, and information about the variance of performance across runs with the same hyperparameters. Compare against other state-of-the-art pruning techniques.

PROBLEM STATEMENT AND PROPOSED SOLUTION

Problem: Winning tickets are currently expensive to find. The best known procedure [1, 2] is "iterative magnitude pruning" (IMP), which involves repeatedly training a network to completion, pruning by a fixed percentage, and "rewinding" weights to an early iteration of training until the network reaches the desired level of sparsity. To reach sufficient sparsity on standard networks, this procedure must be repeated 10 or more times.

Goal: To propose a procedure that finds winning tickets more efficiently.

Significance: A more efficient procedure would make it easier to study the lottery ticket phenomenon. (Whether studying that phenomenon is, itself, significant is debatable.) Personally, I have extensive experience using IMP to find winning tickets, so techniques to reduce the cost of finding winning tickets would be very valuable for my work.

Proposed solution: The authors propose "continuous sparsification" (CS), which makes it possible to learn which weights to prune simultaneously with the weights themselves. Accompanying each parameter w in the network is a second parameter s. The actual weight used in the network is w * sigmoid(s * beta), where beta is a temperature hyperparameter. If the learned value of s is such that sigmoid(s * beta) is approximately 0 at the end of training, then the parameter has been pruned. The value of beta increases exponentially throughout training, meaning the output of the sigmoid will be closer to a hard 0 or 1, producing a pruning mask. To ensure sparsification happens, a regularization term lambda |sigmoid(beta * s)| is added. When run over multiple iterations, values of s are reset to their original values for weights that are not pruned. In addition, weights are either rewound (as in IMP) or left at their final values (as in [3]).

Novelty: The paper is slightly novel. The technique is a variation of those proposed in [4] and [5], but the changes are meaningful. This is the first known use for finding winning tickets, but any pruning technique could hypothetically be used for this purpose. In effect, the paper proposes a new pruning technique that is primarily evaluated for its efficacy in finding winning tickets.

TECHNICAL REVIEW

* This technique introduces several new hyperparameters: lambda, initial and final betas, and the initial values for s. The paper suggests good values for these hyperparameters for both networks considered. These values were presumably found through hyperparameter search of some kind. How extensive was this hyperparameter search, and what is the range of "good" combinations of values? This is not just a methodological footnote; this paper's stated goal is to improve the efficiency of finding winning lottery tickets, and - if good hyperparameters are hard to find - then that defeats the purpose of a more efficient technique. IMP, while far less efficient epoch for epoch as compared to CS, requires no hyperparameter search; global pruning 20% of parameters per iteration seems to work well in general [2]. In a revised version of the paper, I would be eager to learn more about this set of tradeoffs, since that is what matters in practice.

* The authors only study two networks: a toy convolutional network and a small Resnet. It is hard to draw broad conclusions from such a limited set of examples. I would be particularly interested in seeing how this technique performs on a large-scale network for ImageNet (e.g., Resnet-50), since these are the situations where IMP becomes particularly cost-prohibitive. If the technique works well in these settings, it would enable lottery ticket research at much larger scales than is currently possible. If the technique works as efficiently at this scale, then doing could even be feasible during the rebuttal period. (I acknowledge getting experiments working on ImageNet is no small undertaking in terms of both engineering time and cost, but it would improve my confidence to see those results.)

* The authors did an admirably careful job replicating the networks in [1], which include a variety of nonstandard hyperparameters.

* Did the paper study Resnet-18 (a network designed for ImageNet with 11.2M parameters) or Resnet-20 (a network designed for CIFAR-10 with 272K parameters)? Frankle et al. [1, 2] describe Resnet-20 in their appendices but mistakenly refer to it as Resnet-18 throughout both papers, so I wanted to clarify. Based on the final test accuracy of the network, it appears to be Resnet-20; if so, I'd urge you to call it as such and note in a parenthetical or footnote that it's the same network as in [1, 2] but with Frankle et al's mistaken name corrected.

* Are the values of s0 for CS sampled from a distribution, or are they all the same, fixed value?

* Do the extra parameters lead to longer wall-clock training times? If you are making an argument about a more efficient technique, this is an important consideration.

* Figure 2 includes the same graph twice. I believe a different graph should appear on the left, and I am eager to take a look at it in a revised version of the paper.

* It does not appear that multiple replicates of each experiment were run with different network initializations in Figure 3. There don't appear to be any error bars on Figure 3, suggesting that this only represents a single initialization. Considering the wide variance in performance achieved by continuous sparsification across runs in Figure 3 (right), this graph needs to include multiple runs and error bars. (I assume the multiple runs shown in Figure 3 (right) are with different hyperparameters?)

* What is the performance of CS as a pruning technique? By finding winning tickets, CS is also implicitly pruning the network. Is this competitive with L0 Regularization [4]? Is it more efficient than iterative magnitude pruning as in [3]? If this is a more efficient technique for finding winning tickets, it is also liable to be a more efficient pruning method, which seems like even broader impact for the proposed technique. Alternatively, if this technique is less effective than comparable work (especially [4]) as a pruning method, then it is possible comparable work (especially [4]) might also produce better winning tickets, in which case the importance of this work is diminished. In an updated version of the paper, I would be interested in seeing an evaluation of CS as a pruning technique independent of the lottery ticket hypothesis (and compared to standard techniques in the pruning literature as such). I have a hard time seeing any reason why new techniques for finding lottery tickets are any different than new pruning techniques, and they should be evaluated as such in the context of the broader literature on pruning.

* The last paragraph of Section 4.1 is missing important details that are necessary to evaluate the utility of continuous sparsification in comparison to IMP. "In only two iterations, CS finds a ticket with over 77% sparsity..." - for which hyperparameters, and how many hyperparameters had to be explored to find these values? How was the pareto curve obtained? How many different runs were necessary to create it? How many total epochs of training did it take to find that curve? If obtaining this pareto curve required many runs of CS, then it may not be any more efficient than running IMP in practice. The pareto curves appear to make CS look misleadingly effective, since they hide many of the actual costs involved (e.g., hyperparameter search, number of separate runs that were conducted to produce the curve, etc.). Greater transparency of this aspect of the paper would go a long way toward increasing my confidence that the findings are an improvement over IMP.

* The sparsity and accuracy of subnetworks found by CS appears to vary widely from run to run as shown in Figure 3 (right). The final sparsity appears to be a function of the values of s0, lambda, and beta, potentially along with luck from the optimization process. For the same values of s, lambda, and beta, how widely do the final sparsities and accuracies vary? If I wanted to find a winning ticket with a particular sparsity using CS, what would the procedure look like for doing so? Would I have to sweep across values of these hyperparameters, or is there a more straightforward way to do so? The practical usefulness of the procedure hinges on these questions.

* "We associate the performance drop of highly sparse tickets found by our method from the second iteration onwards to the lack of weight rewinding." Why didn't you also try it with weight rewinding? That seems like an easy way to evaluate this hypothesis. For both networks, it would be interesting to see the performance of CS with and without rewinding (analogous to Appendix B in [1]).

* In the literature so far, the only winning tickets to be examined are those from IMP [1, 2]. CS is a different technique, and it likely finds different winning tickets. Do these winning tickets have different properties from [1, 2]? Can they be reinitialized? Can they be rewound earlier? These comparisons seem like an interesting scientific opportunity.

* There are a few additional comparisons that I think are vital to include in the paper to appropriately contextualize results.  They're in the bullets below.

* First comparison: it would be useful to include random reinitialization or random pruning baselines as in [1, 2] simply to make it easier for the reader to contextualize the performance of other sparse subnetworks.

* Second comparison: what happens if you run IMP such that each iteration prunes to the same sparsity as achieved by each iteration of CS? Perhaps pruning by a fixed amount per iteration in IMP is wasteful, and one can prune more aggressively during earlier iterations as CS naturally appears to do. In other words, one way of explaining the advantage of CS would be that it prunes more aggressively. Is this indeed the case? I would be very curious to know.

* Third comparison: What are the range of results achieved if IMP and CS are run "one-shot" (pruning after just one iteration; in the case of IMP, pruning directly to a desired sparsity)? That is, how well can these techniques do with just a single iteration?

WRITING

The writing is excellent. The prose is clear, and I was able to fully understand a relatively sophisticated technique on the first read through the paper. Writing of this quality is rare, and the authors should be commended for it.

OVERALL: Weak Reject

The problem statement is that IMP is not efficient. The paper claims that CS is more efficient. However, the paper does not present a convincing case that CS is, on the whole, more efficient when taking into account hyperparameter search to get CS to work, hyperparameter search to target a particular sparsity, potential variance across runs of CS, potential additional training costs of CS, and the possibility that IMP might be able to work comparable well given a more aggressive pruning schedule.

In addition, the evaluation needs more experiments, including multiple replicates for each experiment and more networks (ideally one on ImageNet).

Finally, I am unclear on what distinguishes CS from any other pruning technique, and it should be evaluated in the context of the broader pruning literature.

If the authors clarify that the overall cost of CS (including all of the factors listed above) is lower than for IMP, address technical concerns about the evaluation, and evaluate CS as a pruning technique in an updated version of the paper, I will update my score accordingly.

[1] Frankle & Carbin. "The Lottery Ticket Hypothesis." ICLR 2019.
[2] Frankle et al. "Stabilizing the Lottery Ticket Hypothesis." Arxiv.
[3] Han et al. "Learning both Weights and Connections for Efficient Neural Networks." NeurIPS 2015.
[4] Louizos et al. "Learning Sparse Neural Networks through L0 Regularization." ICLR 2018
[5] Zhou et al. "Deconstructing Lottery Tickets: Signs, Zeros, and the SuperMask." NeurIPS 2019.

=====================================

COMMENT TO THE AC AFTER READING REBUTTALS AND REVISED PAPER

TLDR

I believe the technique is novel and should be published, regardless of whether it is actually more efficient than IMP for practical workflows.

However, I believe the current evaluation is inadequate, both in the experiments and in the way data is presented. Namely, it is impossible to actually compare the costs of CS and IMP in the scenarios the authors evaluate, despite the fact that efficiency is the authors' claimed contribution. From the data as presented, it is unclear whether CS is actually more efficient than IMP for scientific use-cases. I'm not particularly concerned if there isn't an efficiency advantage - it's an innovative contribution regardless. My concern is that it is impossible to compare the costs of these techniques given the current presentation of data. That speaks to flaws in the evaluation section.

I therefore maintain my score: weak reject. The technique deserves to be published, but the paper in its current form does not.

INTRODUCTION

After reading the original submission, I had the following questions:

1) Right now, the only way to find winning lottery tickets is through training the network and pruning it. IMP can be instantiated with any pruning technique, and the authors are really proposing a new pruning technique for use in the IMP framework. How does continuous sparsification perform as a pruning technique for network compression (i.e., independent of the lottery ticket hypothesis)?

2) How efficient is continuous sparsification (CS) in the scientific use cases where one would seek to find winning lottery tickets? In my research experience, there are two such use cases: (a) producing a winning lottery ticket with a specific sparsity and (b) producing winning lottery tickets across the full range of different sparsities.

In their reubttal, the authors addressed these underlying questions:

AS A PRUNING TECHNIQUE

"We have added new experiments showing that our method yields competitive results when pruning VGG trained on CIFAR, outperforming both magnitude pruning and stochastic l0 regularization"

My response: The initial results that the authors present are quite impressive on a VGG-style network for CIFAR-10. However, I find it concerning that the authors study pruning on this specific model but no others, particularly because they focus on a different network (namely, Resnet-20) in all other experiments in the paper. Since the authors are, in essence, proposing a new pruning technique, I would like to see it comprehensively evaluated as such on a range of networks against other baselines (as the authors do in Figure 4 for one network - those baselines are fine to me).

EFFICIENCY OF CONTINUOUS SPARSIFICATION

- It increases the cost of each individual network training run slightly: "Continuous Sparsification on a 1080 Ti: our method resulted in 15% extra wall-clock time per training epoch."

- PRODUCING WINNING TICKETS ACROSS THE FULL RANGE OF SPARSITIES: Throughout the paper, the authors present "pareto curves" showing the highest accuracy achieved by CS subnetworks at various sparsities. I find these pareto curves misleading: they hide the fact that CS had to be run several times with different hyperparameters to produce these curves.

For example, in Figure 6 (right), the authors produce the pareto curves by training Resnet-20 (by my count) 22 times (each point along the purple lines). The corresponding IMP curve appears to have 14 points. In other words, to get winning lottery tickets across all sparsities, CS must be run for more iterations than IMP - it appears to be less efficient, contradicting the authors' core claim. The best argument in favor of CS is that one could perform each of these runs in parallel if sufficient GPUs were available, meaning less wall-clock time would be required.

One caveat to this analysis: the authors state that they run CS "without rewinding," meaning that the second iteration of CS requires less training time than fully training the network (as IMP requires). The authors do not state how long they train when they *aren't* rewinding, so it is impossible to compare the efficiency of CS with IMP. All they say is that it "allow[s] for even faster ticket search." They also do not study IMP without rewinding, which would be a helpful baseline for comparing to CS without rewinding.

***In short, if the authors are making an argument that one technique is more efficient than the other on an epoch-for-epoch basis, they need to actually plot the epochs required by each technique.***

- PRODUCING A WINNING TICKET AT A SPECIFIC SPARSITY: My concern about this use case is that there is a non-intuitive relationship between the initialization of the sparsity parameters and the final sparsity of the network. As the authors state in the rebuttal: "To achieve a desired sparsity, one can either perform runs in parallel with different values for s_0, or perform sequential binary search if the goal is to minimize the overall computational cost and not wall-clock time." In other words, there is no precise way to target a particular sparsity other than trying many hyperparameter configurations. The authors do not provide any concrete costs of using CS in this way in comparison to IMP, and - from a usability perspective - this is a challenging workflow.

OTHER CONCERNS

- The authors only study CS on one toy network (the six-layer convolutional network, which - in my experience - is a particularly easy setting compared to deeper networks) and one "real" network (Resnet-20 on CIFAR-10). I would like to see results on other networks for CIFAR-10 (for example, the VGG network in Section 4.3).

- More importantly, I would like to see results on an ImageNet network. If CS makes finding winning lottery tickets more efficient as the authors claim, then finding winning tickets efficiently on an ImageNet network should be an excellent demonstration of their contribution. This scenario has stretched IMP to its breaking point, as the authors note.

ARGUMENTS IN FAVOR OF ACCEPTING

+ The authors propose a new pruning technique that appears to improve upon the increasingly popular L0-regularization technique.

+ With the right hyperparameters, the proposed technique makes it possible to find winning lottery tickets more efficiently than existing methods (i.e., IMP with magnitude pruning).

+ The winning tickets found by the proposed technique reach higher accuracy than IMP winning tickets and produce winning tickets at more extreme sparsities, improving upon our knowledge of the existence of winning lottery tickets.

ARGUMENTS IN FAVOR OF REJECTING

- The authors perform only minimal evaluation of their method as a pruning technique. It is possible that this is a missed opportunity to show an additional contribution. It is also possible that other, existing pruning techniques outperform CS at both pruning and finding winning lottery tickets.

- The technique is hard to use for the two existing use cases for finding winning tickets. In particular, there is no way to search for winning tickets at a specific sparsity.

- It is unclear whether CS is actually more efficient than IMP on an epoch-for-epoch basis, even though this is the main claimed contribution. The authors do not disclose - let alone plot - the number of training epochs required to find (a) winning tickets at a specific sparsity and (b) winning tickets at a range of sparsities, so it is impossible to make these comparisons. Meanwhile, the pareto curves the authors present are misleading, since they are the amalgamation of many separate runs of CS.

- The authors study their technique on only one "real" network (Resnet-20) for finding winning tickets and a separate "real" network (VGG-19) for pruning. The authors do not show how CS performs in other challenging settings, especially ImageNet.

- This work is only valuable if we believe that "lottery ticket hypothesis" work is valuable. In other words, this is a narrow contribution to an already-narrow area of study. This is one reason why pitching CS as a pruning technique would make this a stronger paper. I personally believe that "lottery ticket" work is valuable area of study, but I understand that it may not be seen as such in the broader ICLR community.

CONCLUSION

I believe the technique deserves to be published regardless of whether it is actually more efficient than IMP. It is a novel contribution to both our knowledge of pruning and of finding winning lottery tickets. The paper is exceptionally well written, and - as such - I believe it will help to inspire other research in this area.

However, I do not believe that the current paper - namely, the current evaluation - should be published. The paper presents no concrete data on the comparative costs of performing CS and IMP even though the core claim is that CS is more efficient. The paper does not disclose enough detail to compute these costs, and it seems like CS is more expensive than IMP for standard workflows. Moreover, the current presentation of the data through "pareto curves" is misleadingly favorable to CS.

I also believe that the paper needs experiments on ImageNet and needs a more thorough evaluation as a pruning technique beyond the lottery ticket hypothesis.

I therefore retain my current score of "weak reject," though I am eager to hear the thoughts of other reviewers, and I am open to changing my score.


**Experience Assessment:**

I have published one or two papers in this area.

**Review Assessment: Checking Correctness Of Derivations And Theory:**

N/A

**Review Assessment: Checking Correctness Of Experiments:**

I carefully checked the experiments.

**Review Assessment: Thoroughness In Paper Reading:**

I read the paper thoroughly.

---

> ### Author Response · Authors · 2019-11-13
> **Response to reviewer 4 [2/2]**
>
> - “The last paragraph of Section 4.1 is missing important details (...) for which hyperparameters, and how many hyperparameters had to be explored to find these values?”
>
> We have added details on exactly what hyperparameters were used to achieve the results in Figure 3. In particular, we used a fixed final temperature of 250, a penalty lambda of 1e-10, and varied s_0 across 6 different values (-0.05, -0.03, -0.02, -0.01, -0.005, 0) to control the sparsity of the found ticket. We also added a section to the Appendix showing how each hyperparameter affects our method: in a nutshell, our results are robust to changes in both the final temperature and lambda (we used a fixed starting temperature of 1 across all runs), showing that our main hyperparameter is the mask initialization s_0.
>
> ———————————————-
>
> - “The final sparsity appears to be a function of the values of s0, lambda, and beta” / “For the same values of s, lambda, and beta, how widely do the final sparsities and accuracies vary”
>
> Our responses above also address these points: in particular, we have added error bars to Figure 3 to show how the sparsity varies across runs with the same hyperparameter settings; we also added a section to the Appendix studying how each hyperparameter affects our the sparsity of the tickets.
>
> ———————————————-
>
> - “If I wanted to find a winning ticket with a particular sparsity using CS, what would the procedure look like for doing so?
>
> This is an excellent question. As mentioned previously, in practice the sparsity of the final model is almost fully determined by the value of s_0. To achieve a desired sparsity, one can either perform runs in parallel with different values for s_0, or perform sequential binary search if the goal is to minimize the overall computational cost and not wall-clock time (new results in the Appendix show that sparsity is monotonically decreasing with s_0, making binary search possible). In practice, we observed that s_0 = 0 yields high accuracy and around 70% final sparsity for both ResNet-20 and VGG, and s_0 in {-0.01, -0.05, -0.1} is typically enough to achieve well-spaced sparsity values up to 95%.
>
> ———————————————-
>
> - “Why didn't you also try it with weight rewinding?”
>
> We observed empirically that rewinding offers an explicit trade-off between required training time and how good the mask is after many training iterations. More specifically, training with rewinding allows for our method to perform many iterations without the mentioned performance drop — however, training without rewinding increases the performance in the first iterations, since the network does not need to be fully re-trained at each iteration. We will add an extra section to the Appendix with empirical results showing this phenomena to the camera-ready version.
>
> ———————————————-
>
> - “Do these winning tickets have different properties from [1, 2]? Can they be reinitialized? Can they be rewound earlier?”
>
> We have added empirical results showing that tickets found by Continuous Sparsification cannot be re-initialized without significant performance degradation (dashed lines in Figure 3). We also confirmed that rewinding to epoch 2 is necessary to find winning tickets on ResNet 20: disabling it (rewinding to initialization) yields tickets that underperform the original, dense network.
>
> ———————————————-
>
> - “it would be useful to include random reinitialization or random pruning baselines as in [1, 2] “
>
> We have added random initialization for tickets to the revised version of the paper (dashed curves in Figure 3). We will add random pruning baselines to the camera-ready version.
>
> ———————————————-
>
> - “what happens if you run IMP such that each iteration prunes to the same sparsity as achieved by each iteration of CS?”
>
> This is an interesting venue for exploration. In practice, we observe that when the final sparsity is large enough (e.g. over 80%) CS performs virtually all pruning in the first one or two iterations, hence replicating the pruning rate with IMP would be similar to running it for a single iteration with a large pruning rates. We have added results where we run IMP with pruning rates larger than 20% to the Appendix: in particular, there is visible performance degradation of the tickets found by IMP even with a pruning rate of 40%. We will add extra experiments to the camera-ready version, where we run IMP to mimic the pruning rate of CS in each iteration.
>
> ———————————————-
>
> - “What are the range of results achieved if IMP and CS are run "one-shot" (pruning after just one iteration; in the case of IMP, pruning directly to a desired sparsity)?”
>
> We have addressed this point above: we added Section 4.3 where CS is compared against magnitude pruning and stochastic l0 regularization in the task of one-shot pruning on VGG.

---

> ### Author Response · Authors · 2019-11-13
> **Response to reviewer 4 [1/2]**
>
> Thank you for your extensive comments and detailed review. We address your points individually below — please let us know if we can clarify or address any further concerns.
>
>
>
> - “the proposed technique is inconsistent across runs”
>
> Our method is in fact consistent across runs. To clarify, Figure 3 shows both a pareto curve (green) and multiple runs of Continuous Sparsification, each with different hyperparameter settings. For a given set of hyperparameters, the behavior of our method is consistent. The different sparsity trajectories are attained by running our method with different hyperparameter settings (which directly, and intuitively, affect the final sparsity of the model).
>
> We have re-ran our experiments with 3 different random seeds for each hyperparameter setting, and have added error bars to Figure 3 accordingly. Note that the variance of the tickets’ performance from the 2nd round onward (2nd and next markers of purple curves) is smaller than the variance of tickets found by IMP. We have also added an Appendix with hyperparameter analysis for our method, showing that changes in hyperparameter values have consistent impacts on the performance and sparsity of found tickets.
>
> ———————————————-
>
> - “I would be particularly interested in seeing how this technique performs on a large-scale network for ImageNet “
>
> We have not performed these experiments due to the computational costs of fully training an ImageNet model for many iterations in a sequential fashion. In particular, Frankle et al. (“Stabilizing the Lottery Ticket Hypothesis”) train a ResNet 50 for over 1300 epochs on ImageNet. Nonetheless, we are currently working on evaluating our method when training a ResNet 50 on ImageNet, and will add them to the camera-ready version of the paper.
>
> ———————————————-
>
> - “Did the paper study Resnet-18 (a network designed for ImageNet with 11.2M parameters) or Resnet-20”
>
> Thanks for pointing this out. We used a ResNet-20 for our experiments, and we have updated the paper to clarify this.
>
> ———————————————-
>
> - “Are the values of s0 for CS sampled from a distribution, or are they all the same, fixed value?”
>
> We initialize all parameters of the soft mask with the same value. Note that the value used significantly affects the final sparsity of the model: in Figure 3, different runs of CS had all hyperparameters fixed except for s_0. We have added more details on how s_0 was chosen to generate the results in Figure 3, and the new section in the Appendix shows how s_0 affects the sparsity of the tickets.
>
> ———————————————-
>
> - “Do the extra parameters lead to longer wall-clock training times? If you are making an argument about a more efficient technique, this is an important consideration.”
>
> We have measured the wall-clock training time of Iterative Magnitude Pruning and Continuous Sparsification on a 1080 Ti: our method resulted in 15% extra wall-clock time per training epoch. We have added this information to the revised paper.
>
> ———————————————-
>
> - “Figure 2 includes the same graph twice. I believe a different graph should appear on the left, and I am eager to take a look at it in a revised version of the paper.”
>
> Thanks for pointing this out. We have corrected this in the revised version of the paper.
>
> ———————————————-
>
> - “It does not appear that multiple replicates of each experiment were run with different network initializations in Figure 3.”
>
> As mentioned above, we have updated Figure 3 to have error bars computed from runs with 3 different random seeds.
>
> ———————————————-
>
> - “What is the performance of CS as a pruning technique?”
>
> We have added new experiments showing that our method yields competitive results when pruning VGG trained on CIFAR, outperforming both magnitude pruning and stochastic l0 regularization (with straight-through gradient estimation). Results are presented in Section 4.3: our method successfully prunes 99.7% of the parameters while still maintaining over 90% test accuracy, while both magnitude pruning and stochastic sparsification suffer from severe performance degradation when over the pruning rate is over 98%, achieving less than 70% accuracy. We will add results for the stochastic l0-based method in Louizos et al., which uses the hard-concrete distribution, to the camera-ready version of our paper.

---

> ### Comment · AnonReviewer4 · 2019-11-13
> **Will Review Response Soon!**
>
> Thank you to the authors for their detailed response. I'll try to take a look soon and offer more feedback.

---

### Author Response · Authors · 2019-11-13
**Paper revision**

We thank all the reviewers for the valuable feedback. We have revised the paper, incorporating several changes suggested in the reviews. The major changes are:

- Experiments for Figure 3: added error bars computed from 3 different runs, added curves showing performance of tickets when re-initialized (dashed lines).

- Added Section 4.3, where we perform one-shot pruning on a VGG network, and compare against magnitude pruning and stochastic l0 regularization.

- Added an Appendix with empirical analysis on how each hyperparameter affects the performance and sparsity of tickets found by our method.

We will make our code publicly available in the near future.

---

### Decision · Program_Chairs · 2019-12-19

**Decision:**

Reject

**Comment:**

This paper proposes a new algorithm called Continuous Sparsification (CS) to search for winning tickets (in the context of the Lottery Ticket Hypothesis from Frankle & Carbin (2019)), as an alternative to the Iterative Magnitude Pruning (IMP) algorithm proposed therein. CS continuously removes parameters from a network during training, and learns the sub-network's structure with gradient-based methods instead of relying on pruning strategies. The papers shows empirically that CS finds lottery tickets that outperforms the ones learned by ITS with up to 5 times faster search, when measured in number of training epochs.

While this paper presents a novel contribution of pruning and of finding winning lottery tickets and is very well written, there are some concerns raised by the reviewers regarding the current evaluation. The paper presents no concrete data on the comparative costs of performing CS and IMP even though the core claim is that CS is more efficient. The paper does not disclose enough detail to compute these costs, and it seems like CS is more expensive than IMP for standard workflows. Moreover, the current presentation of the data through "pareto curves" is misleadingly favorable to CS. The reviewers suggest including more experiments on ImageNet and  a more thorough evaluation as a pruning technique beyond the lottery ticket hypothesis. We recommend the authors to address the detailed reviewers' comments in an eventual ressubmission.